



# Tectonostratigraphy of the Mérida Massif reveals a new
# suture zone exposure in SW Iberia
**Rubén Díez Fernández[1]\*, Ricardo Arenas[2], Esther Rojo-Pérez[2], Sonia Sánchez**
**Martínez[2], José Manuel Fuenlabrada[3]**
*[1]Departamento de Geodinámica, Estratigrafía y Paleontología, Universidad*
*Complutense de Madrid, 28040 Madrid, Spain*
*[2]Departamento de Mineralogía y Petrología, Universidad Complutense de Madrid,*
*28040 Madrid, Spain*
*[3]Unidad de Geocronología (CAI de Ciencias de la Tierra y Arqueometría), Universidad*
*Complutense de Madrid, 28040 Madrid, Spain*
\*Corresponding author: rudiez@ucm.es
**ABSTRACT**
Dividing a crystalline basement into tectonostratigraphic units, along with the recognition
of the nature of their boundaries (primary vs. tectonic), are essential steps to identify
major tectonic slices involved in orogeny. The Neoproterozoic and Paleozoic rocks of the
Mérida Massif (SW Iberia) have been grouped into five tectonostratigraphic units
according to their structural position, continental or oceanic crust affinity, and equivalent
tectonometamorphic evolution. Each unit is separated from the rest ones by either crustal-
scale thrusts and/or extensional detachments. The lowermost unit (Magdalena Gneisses;
lower plate) has continental crust affinity, and rest below a variably strained and





metamorphosed mafic-ultramafic ensemble, referred to as the Mérida Ophiolite (suture
zone). The Neoproterozoic Montemolín Formation of the Serie Negra Group constitutes
a unit with continental crust affinity (Upper Schist-Metagranitoid Unit; upper plate)
located on top of the Mérida Ophiolite. A carbonate-rich succession (Carija Unit)
occupies the uppermost structural position. Structural and isotopic data suggest that the
suture zone depicted by the Mérida Ophiolite and the tectonic piling and main foliation
of the Neoproterozoic and Cambrian units were formed during the Cadomian Orogeny.
Superimposed shortening during the late Paleozoic formed a train of upright to NE-
verging folds and thrusts that affected the Cadomian suture zone and juxtaposed it onto
Ordovician strata (fifth tectonostratigraphic unit) during the Variscan Orogeny. Cenozoic
contraction during the Alpine Orogeny formed SW-directed thrusts in an intraplate
setting. The Mérida Ophiolite represents a new Cadomian suture zone exposure of the
Iberian Massif, but its root zone is yet to be identified. This suture zone exposure seems
to share a far-travelled nature with other Cadomian and Variscan suture zone exposures
in Iberia, making the latter a piece of continental lithosphere built at the expense of
allochthonous terranes transferred inland from peri-Gondwana onto mainland Gondwana,
both during the Neoproterozoic-Cambrian and the Devonian-Carboniferous.

**Keywords**: Cadomian tectonics; Cadomian allochthons; Cadomian suture; Ophiolite;
Variscan tectonics; SW Iberian Massif

**1. INTRODUCTION**

Bedrock mapping of regions featured by a rich variety of rocks may result in

contrasting outcomes depending on lithological grouping criteria. Establishing coherent
lithological groups to be mapped is essential. Classical criteria to divide crystalline

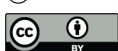



basement bedrock include: (i) structural position at regional scale, (ii) continental,
oceanic or transitional crust affinity, and (iii) equivalent tectonometamorphic evolution.
Such approach leads to the establishment of what is usually referred to as the
tectonostratigraphy of a region, a practical concept that gathers compositionally complex
rock sequences layered after penetrative deformation affecting one or more sections of
the lithosphere, each of which would represent an independent tectonostratigraphic unit.
This workflow has advantages and drawbacks of its own, but allows identifying major
geological features and helps set the base for future, more oriented, research. Grouping
of contrasting lithologies makes detailed time-resolved geodynamic reconstructions
based on petrological aspects more difficult, but on the other hand it favors recognition
of terranes (e.g., major tectonic blocks) by acknowledging their internal complexity. This
purpose-oriented method is particularly useful to identify continental blocks intervening
in a suture zone, and is actually the basis for widely accepted ideas in tectonic
reconstructions, such as the ophiolite concept.
Here we present a new simplified geological map and cross-section of the Mérida
Massif aimed to distinguish its main tectonostratigraphic units and their relationships.
Lithological ensembles for each unit have been grouped according to criteria cited above.
The resulting tectonostratigraphy provides light into a nappe structure that had remained
unnoticed for the region, despite large scale thrusting events in SW Iberia have been
claimed for Variscan models (Azor et al., 1994; Díez Fernández and Arenas, 2015;
Ribeiro et al., 2010) and Cadomian tectonics (Díez Fernández et al., 2019; Abalos et al.,
1991). It also adds another ophiolite and suture zone exposure to the set that features the
Iberian Massif, thus strengthening the notion of the Iberian domain as a region built after
numerous suturing processes throughout geological history.



## 2. GEOLOGICAL SETTING


The Iberian Massif constitutes the southernmost exposure of the Variscan Orogen
in Europe (Fig. 1), which resulted from the progressive collision of Gondwana, Laurussia
and their respective pericontinental terranes during the Devonian and Carboniferous
(Simancas et al., 2013; Martínez Catalán et al., 2009; Ribeiro et al., 2007; Díez Fernández
et al., 2016). Yet, the current structure of the Iberian Massif is the result of three orogenic
cycles. Gondwana was affected by long-lived subduction under its periphery during the
Neoproterozoic and Lower Paleozoic (Linnemann et al., 2007; Quesada, 1990; Pereira et
al., 2007; Eguíluz et al., 2000; D'Lemos et al., 1990; Nance et al., 1991; Chantraine et al.,
2001), its northern paleomargin bearing abundant evidence of arc-related magmatism
(Bandrés et al., 2004; Henriques et al., 2015; Dorr et al., 2002; Drost et al., 2004; Rubio-
Ordóñez et al., 2015), basin development in active settings (Rojo-Pérez et al., 2019;
Fuenlabrada et al., 2012; Fuenlabrada et al., 2016; Linnemann et al., 2000; Linnemann et
al., 2007; Fernández-Suárez et al., 2013; Pereira, 2015), and contractional and extensional
deformation (Díez Fernández et al., 2019; Expósito et al., 2003; Simancas et al., 2004;
Eguíluz et al., 2000; Bandres et al., 2002; Kröner et al., 2000; Strachan and Taylor, 1990;
Díaz García, 2006; Balé and Brun, 1989), all of which are collectively referred to as
Cadomian Orogeny (cycle). It is well-established that the external section of Gondwana
facing such subduction was involved in the Variscan cycle, whose onset and culmination
may correspond to an extensional event that led to the opening of oceanic basins (e.g.,
Rheic Ocean; Nance et al., 2010; Linnemann et al., 2007), and the raise of the Variscan
Orogen after their suturing (Matte, 1991; Martínez Catalán et al., 2009; Ballèvre et al.,
2009; Franke, 2000), respectively. The Iberian Massif contains Cenozoic mountain
ranges formed during the Alpine cycle (e.g., de Vicente and Vegas, 2009). Some of them
occur at the boundaries of the Iberian micro-plate (e.g., Pyrenees, Betics), while others



occupy intra-plate positions. Both are the result of plate tectonics in the Mediterranean
domain as well as of distributed strain upon Africa-Europe convergence (Dewey et al.,
1989; Jolivet et al., 2008; de Vicente et al., 2018).

The Mérida Massif is located in the SW part of the Iberian Massif (Fig. 2).

Previous studies have suggested that its composition and current structure is the result of
Cadomian, Variscan, and Alpine tectonics (Bandrés, 2001; Gonzalo, 1987, 1989; Insúa
Márquez et al., 2003). This massif includes an extensive exposure of Neoproterozoic and
Lower Paleozoic rocks (Fig. 3), and represents a good opportunity to study Cadomian
tectonics and the interference of subsequent orogenic cycles over Cadomian imprint.
Mapping of its bedrock geology has provided rather different outcomes over the years
(Insúa Márquez et al., 2003; Bandrés, 2001; Gonzalo, 1987; Roso de Luna and Hernández
Pacheco, 1950). Although poorness of exposure may explain some variation, most of it
seems to derive from contrasting criteria followed during mapping. The Mérida Massif,
although relatively small in size, is characterized by a significantly rich variety of rocks,
making it difficult to establish coherent lithological groups to be mapped if not oriented
to a purpose. Deformation in this area includes the development of foliations, folds, faults,
and shear zones, some of which are coeval to pervasive metamorphism (Gonzalo, 1987,
1989; Bandrés, 2001; Bandrés et al., 2000).

**3. TECTONOSTRATIGRAPHY**

In this section, we will provide a brief description of the main lithological

associations we have established in the Mérida Massif. Grouping is aimed to the
recognition of major tectonic blocks intervening in Cadomian and Variscan tectonics.
Rocks included into each tectonostratigraphic unit meet the following grouping criteria,
and they are different from those gathered into other units for the same reasons: (i) similar

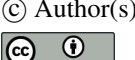



structural position at regional scale, (ii) the ensemble shows either continental or oceanic
crust affinity, (iii) equivalent tectonometamorphic evolution, and (iv) they are separated
from other units by either a major mechanical boundary (i.e., fault or ductile shear zone)
or a major stratigraphic discontinuity (e.g., discordance). Age of protoliths alone was not
taken as a grouping criteria, because the amalgamation of two tectonic blocks may
juxtapose rocks of similar age located at each block. Descriptions are given following
reverse chronological order according to their (meta-) sedimentary strata. In case bedding
and age constrains are lacking, overlying units (as indicated by their main foliation) will
be described first.

**3.1. Cenozoic cover**

The crystalline basement of the Mérida Massif is discordantly covered by a wide

variety of Cenozoic sedimentary rocks. Since Cenozoic processes are not the focus of this
contribution, no distinction has been made in the geological map between sequences of
different age and composition. Previous works have divided this Cenozoic cover into
informal units according to their age (Miocene through to Holocene), location, and
sedimentary environment (Insúa Márquez et al., 2003).

The oldest Cenozoic deposit is represented by Miocene conglomerates and arkosic

sandstones, followed by sandstones and conglomerates, and then red silt and clay and
minor sandstones. This continental series is succeeded by Miocene arkosic sandstones,
which are then covered by a Pliocene-Pleistocene succession of conglomerates,
sandstones and silt. Other Pleistocene and Holocene deposits include carbonated deposits
(caliche), carbonated crusts, glacis deposits (irregular pebbles, gravel, and minor silt),
fluvial terraces (rounded conglomerates, sandstones, and silt), alluvial fans, and aeolian
sands.






### 3.2. Ordovician strata


The youngest sedimentary series that is now exposed as metamorphic rocks in the
study area corresponds to a succession of meta-sandstones, meta-conglomerates,
quartzites, and slates that occurs to the north of the Mérida Massif (Fig. 3). The series is
cut by a major fault, so its basal section is not exposed in the study area. The lower part
exposed consists of coarse-grained meta-sandstones (quartz-rich micro-conglomerates),
which are covered by quartzites and orthoquartzites that alternate with slates. Quartzite
beds are thinner upwards while slates become more abundant. This part of the series has
been ascribed to the Early Ordovician (Tremadocian-Floian), and is considered a SW
Iberian correlative to the (Arenig) Armorican quartzite (Insúa Márquez et al., 2003;
Gutiérrez-Marco et al., 2002). The series culminates with black slates and minor layers
of black quartzites exposed within, whose age has been considered to be Middle
Ordovician (Llanvirn-Llandeil) (Insúa Márquez et al., 2003).

### 3.3. Cambrian carbonate-rich series: Carija Unit


This unit gathers a carbonate-rich, meta-sedimentary succession of variably
strained rocks that is exposed northwest of Merida town (Fig. 3), around the Carija hill.
Strain is particularly concentrated along its basal boundary and decreases progressively
upwards. The lower part consists of black calc-schists that rest below a series in which
fine-grained, banded grey-white marbles alternate with fine-grained dark grey-light grey
dolomitic marbles and yellow-brown marbles. This series has been ascribed to the Early
Cambrian (Insúa Márquez et al., 2003), and considered correlative to other carbonate-rich
Early Cambrian successions of SW Iberia (e.g., Sánchez-García et al., 2010).



### 3.4. Upper Schist-Metagranitoid Unit: Serie Negra Group (Montemolín Formation)


This unit comprises metasedimentary and metaigneous rocks, in which mafic,
intermediate, and felsic terms can be recognized. The metasedimentary series includes
schists, grey metagreywackes, black quartzites, and black schists. Black schists and
quartzites occur as decimeter- to meter-scale lenses within the other schists and
metagreywackes, while the latter are featured by cm- to mm-scale compositional layering
that alternates finer and coarser grained rocks (relict of sedimentary bedding).
Metabasites can be found as fine-grained, lens-shaped (meter-scale) bodies dispersed
over the metasedimentary series. Former regional studies identified this series as a
correlative to a section of the Serie Negra Group (Bandrés, 2001). Its primary immature
nature (as inferred from greywackic composition), along with its content in mafic rocks,
suggest it could be equivalent to the lower (and older) member of such group, regionally
referred to as Montemolín Formation (Eguíluz, 1987).
The sedimentary series was the host to a compositionally complex variety of
intrusive igneous rocks. The (finer-grained) metabasites cited above can be distinguished
from metagrabbros that occur as coarser-grained rocks in larger patches and usually next
to metatonalites. Some metagabbros preserve a primary (equigranular) igneous texture.
Differences in grain size between the latter and fine-grained metabasites are not observed
in zones accumulating larger strain. Therefore, current differences in grain size may be
related to its primary texture (basalts/microgabbros vs. gabbros). Metatonalites,
metagranodiorites, and metagranites occur as variably strained, kilometer-scale bodies
within the metasedimentary rock series. Sections of these bodies accumulating more
strain can be observed as mafic, intermediate, or felsic orthogneisses, respectively,
whereas poorly strained sections show good exposures to analyze the primary texture of
their protoliths. Introducing details on the range of primary textures exceeds the purpose



of this contribution, but as a preliminary approach, all metagranitoids presented phaneritic
texture, ranging between fine- and coarse-grained, and showed either equigranular or
porphyritic terms (more common in felsic granitoids). Varied combinations of these
primary end-members plus heterogeneous strain explain the whole microstructural
variety observed in the metagranitoids of this unit, in which it can be recognized felsic
through to mafic metagranitoids with vaguely-defined planar fabric up to gneisses with
well-developed compositional banding or even augen (K-Feldspar) structure.

Being the most heterogeneous in lithological composition, this unit gathers

variably strained lithologies that resulted from metamorphic transformation of a rock
ensemble that included sedimentary and intrusive igneous rocks. None of the lithologies
listed above is separated from the rest ones within this unit by mechanical contacts with
crustal bearing, since no juxtaposition of lithologies with contrasting tectonothermal
evolution (see below) is observed. Therefore the whole ensemble represents a coherent
tectonic slice with continental affinity.

**3.5. Mafic-ultramafic Unit: Mérida Ophiolite**

The Mérida Massif contains an exposure of mafic and ultramafic rocks that have

been grouped into a single tectonostratigraphic unit due to its contrasting composition
and consistent structural position (see below) relative to surrounding lithologies. This unit
contains coarse-grained gabbros, pegmatitic gabbros, coarse-grained metagabbros,
amphibolites, garnet-bearing amphibolites, hornblendites, and serpentinites. It lacks of
felsic and intermediate igneous rocks, and of paraderived lithologies. No cross-cutting
relationships have been observed between pristine igneous rocks and the rest of
metamorphic rocks of this unit. In fact, some of the latter have been directly observed as
the result of variably strained sections of the first along discrete shear zones (e.g.





amphibolites after gabbros). Therefore the whole ensemble is considered as a
heterogeneously strained and variably recrystallized mafic-ultramafic complex.
Ultramafic rocks, such as strongly foliated serpentinites, occur at different levels
across structure within this unit. Poorly-strained gabbros allow inferring primary igneous
textures (will not be described here), none of which dominates over a particular structural
position across this unit. This mafic-ultramafic Unit shows outstanding ocean crust
affinity, and probably represents tectonic slices of lower oceanic crust and upper mantle
that will be collectively referred to as the Mérida Ophiolite.

**3.6. Lower Gneiss Unit: Magdalena Gneisses**
This unit consists of penetratively deformed and strongly recrystallized
metamorphic rocks. A preliminary study does not allow us to recognize between ortho-
and paraderived terms in it, being felsic rocks in all cases. They contain abundant quartz
and feldspar, although abundance in mica varies from one field exposure to the other.
Grain size is usually small (larger crystals in the matrix are 1-2 mm in size). Some
gneisses show lenticular grains that are slightly larger (2-3 mm) than the minerals in the
quarzt-feldspathic matrix, whereas others contain mineral aggregates (mostly made of
quartz and feldspar) that show similar structure. The augen appearance for single-mineral
lenses could derive from a micro-porphyritic texture in the protolith, which could
tentatively be identified as igneous (granitoid?) in nature. Some of these gneisses include
green amphibole and biotite. Mica-rich and augen-free varieties in these gneisses are less
abundant and could represent paraderived rocks.
No mafic rocks were observed within these gneisses, as opposed to the rest of
units around. The absence of mafic rocks makes it difficult to interpret this ensemble as
a piece of either oceanic or transitional crust. Potential protoliths for the lithologies



grouped into this unit (sedimentary and felsic igneous rocks) represent typical
counterparts of continental crust *sensu lato*, although alternative options cannot be ruled
out due to the limited exposure of this unit.

**4. REGIONAL STRUCTURE AND METAMORPHISM**
The youngest regional structure of the Mérida Massif is a set of NE-dipping, high-
angle faults that cuts across the contacts of the Miocene sedimentary rocks (RP-1; Fig. 3)
and juxtaposes the crystalline basement onto some sections of the Cenozoic cover.
Tectonic transport is consistently to the SW. Displacement for all these thrusts, as
deduced from offsets in pre-fault lithological contacts, is quite limited, probably in the
range of tens of meters at most. Some of the sinuous trace of the basal contact of the
Cenozoic cover near these thrusts could be explained by very open folds, e.g. fault-
propagation folds, although no systematic measurement of bedding is available to prove
it. Besides such direct evidence of Alpine tectonics, it should be noted that most of the
exposure of the Mérida Massif occurs in a small peneplain that stands about 30-60 meters
above the areas located to the SW, S, and SE of the Guadiana river, whose trace in the
surroundings of Mérida town could be controlled by some other Alpine thrusts with
similar displacement (vertical offset at 30-60 meters) and kinematics (RP-2; Fig. 3)
(Vegas et al., 2012). The carbonated crusts, proposed to be Pleistocene-Holocene in age
(Insúa Márquez et al., 2003), affects both the Miocene deposits and the overriding
crystalline basement, thus suggesting a post-Miocene and pre-Pleistocene age (probably
Pliocene) for some Alpine thrusts in the region.
The Ordovician and pre-Ordovician rocks of the study area are separated by a SW-
dipping fault, the San Pedro thrust (RP-3; Fig. 3). Kinematics of this fault is top-to-the-
NE, and includes a left-lateral component that makes it an oblique-slip thrust (Gonzalo,



1987, 1989; Bandrés, 2001). Each block of this fault shows slightly different structural
record. The internal structure of the footwall is dominated by Ordovician strata affected
by NW-SE trending, upright to NE-verging overturned folds (Cornalvo synform), to
which local, and single, main foliation is axial planar (Fig. 4a). There, strata is duplicated
and folds are cut by SW-dipping thrusts with moderate offset, which probably represent
minor fault imbricates of the San Pedro thrust. The internal structure of the hanging wall
to the San Pedro thrust is defined by rocks showing a main foliation affected by NW-SE
trending, upright to either NE- or SW-verging overturned folds (Figs. 3 and 4a). SW-
verging folds are scarcer and tend to occur around a NE-dipping thrust that cuts across
the internally folded structure of the upper block to the San Pedro fault, which will be
referred to as the Barranca back-thrust (RP-4; Fig. 3). The main foliation usually runs
parallel to the boundaries of the bodies of metagranitoids that can be observed in the
Upper Schist-Metagranitoid Unit. The Lower Gneiss Unit crops out in the core of a dome-
like fold, the Magdalena antiform or dome (RP-5; Fig. 3), while the Cambrian marbles
occupy the core of a synform paired with the Magdalena antiform, here referred to as the
Carija synform (RP-6; Fig. 3). Using the main foliation as reference, the structure of the
pre-Ordovician units would consist of a tectonic pile with the Lower Gneiss Unit resting
below the Mérida Ophiolite, which would be covered by the Schist-Metagranitoid Unit
(Ediacaran Serie Negra Group) and then the Carija Unit (Cambrian marbles) (cross-
section in Fig. 4b).

The traces of the contacts between tectonostratigraphic units in the hanging wall

to the San Pedro thrust are featured by mylonites. As a preliminary description, the
contact between the Upper Schist-Metagranitoid Unit and the Mérida Ophiolite is defined
by a ductile shear zone that includes a shear band with mylonites (after mafic and
ultramafic rocks, metasedimentary rocks and metagranites) located at the boundary



between tectonostratigraphic units (core of a major shear zone), and a set of variably
strained rocks towards more distal sections (Trujillanos detachment). Kinematic criteria
indicate consistent top-to-the-SSE shear sense. The contact between the Mérida Ophiolite
and the Lower Gneiss Unit is also featured by mylonites (Magdalena thrust). A
preliminary kinematic analysis of this contact provided no consistent results, probably
due to a more complex nature compared to that of other major boundaries in the region
(see discussion section). The contact between the Upper Schist-Metagranitoid Unit and
the Carija Unit is marked by mylonites after limestones and siliciclastic rocks (black calc-
schists) (Carija detachment). Kinematic criteria indicate a consistent top-to-the-NNW
sense of shear. The trace of all of these contacts shows sinuous pattern and run roughly
parallel to the main foliation observed in the hanging wall to the San Pedro thrust, both
(foliation and mechanical contacts) defining the same NE-SW trending folds (Fig. 4a).

Strain and metamorphic recrystallization is heterogeneous, but the metamorphic

grade varies vertically across structure. The mineral assemblages of the main foliation in
the hanging wall to the San Pedro thrust defines an overall normal metamorphic gradient.
As a reference, the main foliation in the metasedimentary rocks of the Upper Schist-
Metagranitoid Unit may include quartz, plagioclase, white mica, biotite, and minor
(secondary?) chlorite, which make a typical greenschist facies fabric (probably in the
biotite zone). The main foliation in the mafic rocks of that unit includes plagioclase, fine-
grained green amphibole, zoisite, epidote, titanite and chlorite, an assemblage also
compatible with greenschist facies conditions. The main foliation in the mafic rocks of
the Mérida Ophiolite is defined by plagioclase, brown-green amphibole (hornblende),
titanite, opaques, and minor rutile. Some exposures include large garnet porphyroblasts,
and altogether define a typical assemblage for amphibolite facies conditions (grain-size
for the meta-mafic rocks in this unit is significantly larger than in the Upper Schist-





Metagranitoid Unit). PT conditions for this assemblage were estimated at 1.2 GPa and
750 °C (Bandrés et al., 2000). The main foliation in the Lower Gneiss Unit includes
recrystallized plagioclase, K-feldspar, green amphibole, biotite and quartz ribbons with
granoblastic polygonal texture. This fabric is occasionally accompanied by patches of
melt crystallized along bands parallel to the main foliation. The regional normal gradient
in the hanging wall to the San Pedro thrust is juxtaposed onto Ordovician strata, which
show a penetrative slaty cleavage formed by quartz, white mica, chlorite, sericite and
opaques (chlorite zone).

**5. DISCUSSION AND PRELIMINARY CONCLUSIONS**
Alpine deformation in the Mérida Massif is limited and mostly restricted to high-
angle thrusts that reworked its crystalline basement and faulted its Cenozoic cover (at
least during the Pliocene). Main shortening direction is NE-SW and dominant tectonic
transport for thrusts is to the SW. Given the distal position to Cenozoic plate boundaries,
this deformation can be framed into an intra-plate setting for the Iberian micro-plate (e.g.,
de Vicente and Vegas, 2009).
The NW-SE trending folds that affect the basement of the Mérida Massif represent
the first deformation for the Ordovician strata, but are at least the second pulse of
deformation taken by the entire set of pre-Ordovician rocks (note their main foliation and
ductile shear zones are affected by these folds). This indicates that these folds, and the
faults they are cut by (San Pedro and Barranca thrusts) are Variscan in age, and that the
main foliation and ductile shear zones in the pre-Ordovician rocks are probably pre-
Variscan and responsible for the layered structure of most of the tectonostratigraphic
units. This pre-Variscan age of deformation is also supported by Sm-Nd dating of
metamorphic garnet growth in the metabasites of the Mérida Ophiolite (555 Ma; Bandrés



et al., 2004). As a consequence, the current contacts between tectonostratigraphic units in
the hanging wall to the San Pedro thrust could be framed in a Cadomian setting. Normal
metamorphic gradient within and around their juxtaposed tectonic blocks, together with
their crustal-scale bearing (they juxtaposed sections with different metamorphic imprint),
suggest they represent large-scale extensional shear zones. The current folded structure
of these shear zones favors a primary flat-lying geometry for all of them, so they should
be referred to as extensional detachments. The Carija detachment affects Cambrian strata,
so the functioning of some of them may be restricted to the Cambrian and perhaps
Ordovician.

The current regional boundaries between all of the pre-Ordovician

lithostratigraphic units are mechanical in nature, so there is no reason to consider the
entire set of basement rocks as a single tectonic block prior to Ordovician times. Some of
these contacts reflected the functioning of rather different faults through time, what may
explain part of their kinematic complexity observed in a preliminary analysis. The Mafic-
Ultramafic Unit of the Mérida Massif separates two lithological ensembles that show
continental crust affinity, the Mérida Ophiolite being an exposure of a suture zone. Should
the current upper boundary of the Mérida Ophiolite be a pre-Ordovician extensional fault
(Trujillanos detachment), the suture zone this mafic-ultramafic unit represents must be
Cadomian in age. The current juxtaposition of tectonostratigraphic units in the Mérida
Massif suggests a large-scale nappe structure for this suture, where the Lower Gneiss Unit
would represent the lower plate, and the Mérida Ophiolite would account for oceanic
lithosphere located at the base of the upper plate, here represented by the Upper Schist-
Metagranitoid Unit. This way, the primary upper and lower boundaries of the Mérida
Ophiolite could be Cadomian accretionary thrusts, the current nappe structure being the
result of a late Cadomian extensional event that operated over previously thickened crust



built at the expense of basal tectonic accretion (peak metamorphic conditions for the
Mérida Ophiolite suggest lower crust depth). The root of the suture zone represented by
the Mérida Ophiolite is not exposed in the study area, as its lower plate (Lower Gneiss
Unit) occurs in a tectonic window. This supports a pre-Variscan low-dipping geometry
for the thrust sheets involved in the suture zone (e.g. Magdalena thrust; Fig. 4b), some of
the tectonostratigraphic units being actual allochthonous terranes.

The sequence and broad timing of tectonic events recognized for the building of

Mérida Massif fit well into the regional geology of SW Iberia. The Cadomian suture zone
and nappe structure identified in the Mérida Massif add to the evidence of Cadomian
tectonics in southern Europe (Eguíluz et al., 2001; Díez Fernández et al., 2019; Quesada,
1990; Simancas et al., 2004; Pereira et al., 2012; Díaz García, 2006; Pieren et al., 1987;
Bandres et al., 2002), which is tightly connected to subduction-accretion processes in the
periphery of the African margin of Gondwana (Orejana et al., 2015; Fuenlabrada et al.,
2012; Rojo-Pérez et al., 2019; Arenas et al., 2018; Sánchez Lorda et al., 2014; Linnemann
et al., 2008; Bandrés et al., 2004). Variscan deformation in the Mérida Massif reworked
but did not reactivate major Cadomian structures such as accretionary faults and
extensional detachments, which are now observed as folded planes cut at high-angle by
Variscan major faults. In this regard, late Cadomian extension can be observed as a
transitional stage to the Variscan cycle, but most importantly, as a contributor to the
stabilization (cratonization) of orogenic crust before subsequent deformation.

The gathering of contrasting individual lithologies into tectonostratigraphic units

is proven here as a powerful tool to identify major tectonic blocks in orogeny, even at the
lithosphere scale. Equivalent approaches performed in other areas of the Iberian Massif
before focused mostly on the identification of major Variscan thrust sheets (Ribeiro et al.,
2010; Ries and Shackleton, 1971; Díez Fernández and Arenas, 2015; Arenas et al., 1986).



Our work presents a case example that proves this method right for Cadomian tectonics,
and adds evidence to the notion that the lithosphere of the Iberian micro-plate was
constructed not only by the functioning of large-scale accretionary faults and tectonic
transport of allochthonous terranes during the Variscan Orogeny (e.g., Martínez Catalán
et al., 2009; Díez Fernández et al., 2016; Ribeiro et al., 2007), but also during the
Cadomian Orogeny (e.g., Díez Fernández et al., 2019; Abalos et al., 1991).

**6. DATA AVAILABILITY**
The data is directly accessible through the published text and figures.

**7. AUTHOR CONTRIBUTION**
RDF, RA and ERP designed the mapping campaign and carried it out. RDF
delineated the maps (Figures 2 and 3) and draw the cross-sections (Figure 4). All authors
discussed and interpreted the data and prepared the manuscript.

**8. COMPETING INTERESTS**
The authors declare that they have no conflict of interest.

**9. ACKNOWLEDGMENTS**
Research funded by Spanish project CGL2016-76438-P (Ministerio de Economía,
Industria y Competitividad).

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

**FIGURE CAPTION**
Figure 1: Zonation of the Variscan Orogen after Díez Fernández and Arenas (2015).
Location of the study area is indicated. Location of map in Figure 2 is indicated.
Figure 2: Regional geological map of the Obejo-Valsequillo Domain (Díez Fernández et
al., in press) showing the location of the Mérida Massif. Location of map in Figure 3 is
indicated.



Figure 3: Geological map showing the distribution of tectonostratigraphic units of the
Mérida Massif.
Figure 4: Cross sections showing the current structure of the tectonostratigraphic units of
the Mérida Massif. (a) Cross section normal to the trace of Variscan folds and faults. (b)
Cross section subparallel to shear direction of Cadomian shear zones and showing the
Cadomian nappe pile of the Mérida Massif. Cenozoic cover is not represented.





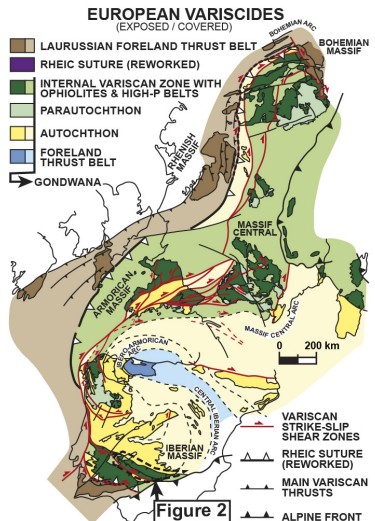

Figure 1





Figure 2

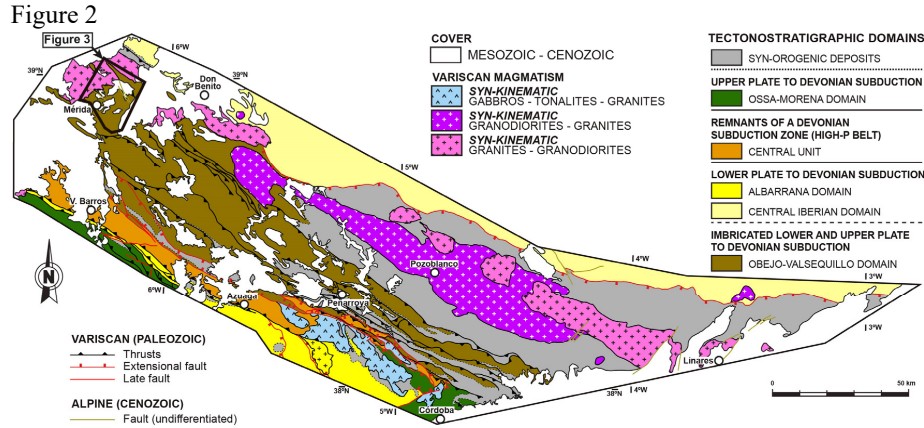



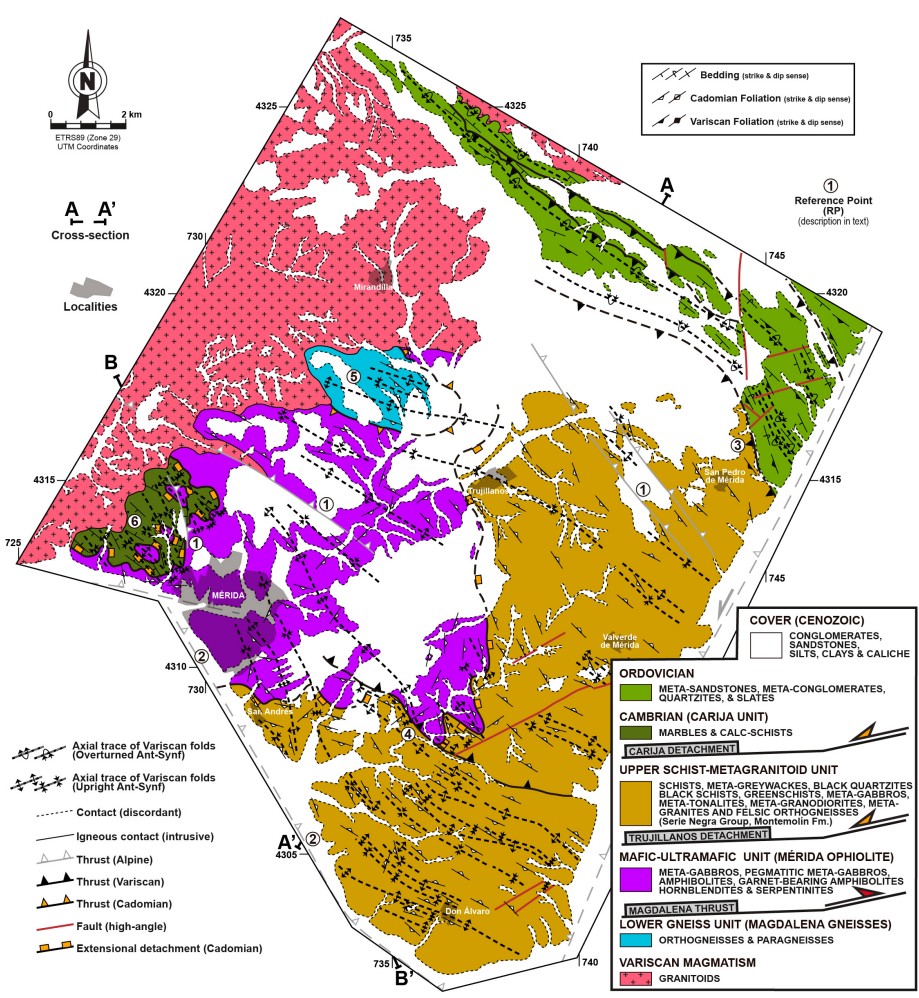

Figure 3



Figure 4

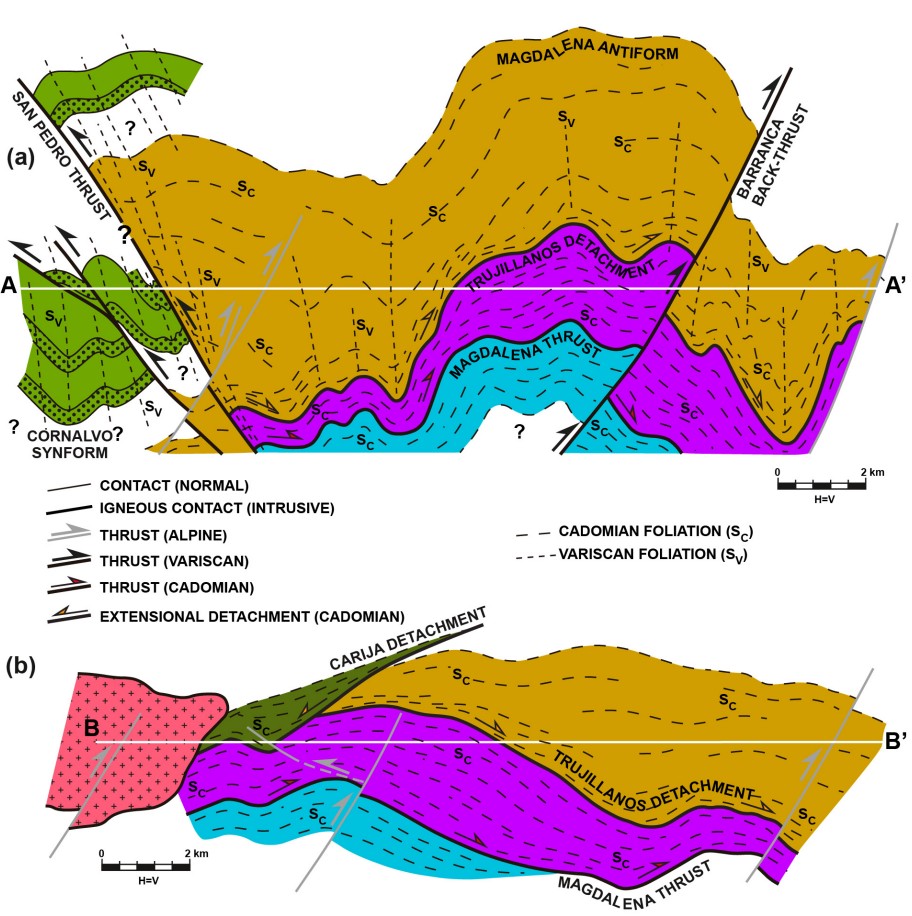