# Peer review of "Tectonostratigraphy of the Mérida Massif reveals a new"

_Solid Earth, 2019_

## Referee Comment (RC1) · Anonymous Referee #1 · 14 Jan 2020

This manuscript presents a new hypothesis on the significance of the gabbro-diorite massif of Merida (Ossa-Morena Zone, SW Iberian Massif), which is interpreted in terms of an ophiolitic complex of oceanic affinity and would attest to a Cadomian suture zone.

Following the initial sections on introduction, general comments and a summary on the available knowledge for the area, the authors briefly describe a number of supposed tectonostratigraphic units, one of which, the so-called Merida Ophiolite, is key to the suture zone hypothesis.

Yet, this hypothesis and the ascription of the Merida massif to an oceanic ophiolitic complex are not supported by any petrological, structural or geochemical data, which

totally compromises the proposed interpretation.

To illustrate the hypothetical tectonostratigraphy, a simplified map of the region is presented, which is inconsistent with published geological maps for this area without the authors justifying theirs.

The authors must provide the pertinent evidence for their assumptions or refer to publications where these evidence are shown. In the absence of such information, it is impossible for any reader to have the smallest idea of the validity of the hypothesis put forward.

---

## Author Comment (AC1) · 15 Jan 2020

Referee's comment #1: "Following the initial sections on introduction, general comments and a summary on the available knowledge for the area, the authors briefly describe a number of supposed tectonostratigraphic units, one of which, the so-called Merida Ophiolite, is key to the suture zone hypothesis."

Authors' reply #1: The term "supposed" gives an idea of the biased nature of the comments that follow. In a way the reviewer insinuates that what we propose is not based on our own data. We remind the reviewer that the map and cross-section are of our own, were created on extensive fieldwork grounds, and are not like others published

before because we used different criteria for grouping rock types. Implying that we have not done fieldwork and own observations in the area and that we base our proposal exclusively on what others published before is unnecessary and disrespectful. We interpret the several groups of lithologies as different tectonostratigraphic units. But it is a fact that one can define a group of lithologies based on whatever he/she likes. The question is if such grouping means something in geological terms. We think the grouping we propose reflects different tectonostratigraphic units, each of which accounting for rather different sections of the lithosphere (and that would be the interpretation/conclusion, not the grouping itself, which is the way to reach such interpretation/conclusion). We do not "suppose" that this would be the most appropriate grouping in general terms. We propose a specific grouping aimed to a purpose, as stated in the manuscript.

Referee's comment #2: "Yet, this hypothesis and the ascription of the Merida massif to an oceanic ophiolitic complex are not supported by any petrological, structural or geochemical data, which totally compromises the proposed interpretation."

Authors' reply #2: As structural data is concerned, foliation/bedding measurements show the relative position of the lithological groups presented in the geological map. Based on that, the mafic-ultramafic complex with ocean lithosphere affinity separates two other units with continental crust affinity. It is very simple structural geology. The structural position of the mafic-ultramafic complex speaks for itself. Additionally, we say (and will show in pictures) that each unit is separated from the rest by a large-scale mechanical contact, which is also typical for the underlying and overlying (if any) units relative to ophiolite complexes exposed inland. Petrological evidence is presented concisely, firstly as a list of lithologies that make the Mérida Ophiolite (and the rest of the units). The lithologies in that list are very common for ophiolites worldwide, which show variety of lithological ensembles depending on the origin of their protoliths, and subsequent evolution. We are dealing with hundreds of meters of mafic rocks (amphibolites and meta-gabbros-diorites of different types) intercalated with metaperidotites

(now serpentinites). That set of rocks does not represent a typical continental litho-sphere. And secondly, the rather different metamorphism experienced by the rocks of the Mérida Ophiolite compared to that of the, for instance, the overlying units, is another evidence on the lithosphere bearing of the mechanical contacts and also on the different metamorphic evolution each of the sections exposed in the Mérida Massif has experienced. This is also very common in suture zones featured by ophiolites, where each terrane (e.g., upper or lower plate) shows a different evolution compared to the ones that are currently juxtaposed with. As geochemical data is concerned, the truth is that this type of data would not test positive the existence of a suture zone by themselves. Among other things, the geochemistry of a rock may give us an idea of the petrological processes and regional (geodynamic) setting where it was formed, but considered alone it says nothing about the oceanic (ophiolitic) or non-oceanic (non-ophiolitic) nature of it. It is the regional context that should be used to identify an ophiolite. The ophiolitic nature of a rock ensemble is something that should be re-solved (or proposed) first, as we are doing in this contribution, based on other grounds (rock associations, structural position relative to other rock associations, comparative metamorphic evolution, nature of lithological contacts, etc.).

Referee's comment #3: "To illustrate the hypothetical tectonostratigraphy, a simplified map of the region is presented, which is inconsistent with published geological maps for this area without the authors justifying theirs."

Authors' reply #3: We wonder if the authors of previously published geological maps justified theirs when the maps where published. And if so, how did they do it? In the revised manuscript, maybe we should include (and we will) some pictures to illustrate the statements we make regarding the mechanical nature of the contacts and the exis-tence of some rocks and structures in particular. But in order to justify a geological map (i.e., the distribution of the groups of rocks we propose) we should provide hundreds of georeferenced and oriented pictures, at least one for each of the outcrops we have visited during mapping. We believe this is something that has never been done in the

history of geology. Maybe the referee is trying to take the publishing of geological maps to a next, more demanding level. Yes, we know, the map does not match with the maps others did before. But that does not mean we are not right, does it? Just for the sake of discussion, it is surprising that some or most contacts shown in previous maps do not parallel the foliation or bedding strike in the region (according to our measurements). And it is also surprising none of the former authors recognized the mechanical nature of some contacts, even after calculating the peak metamorphic conditions for some rock exposures. It seems the study area is a box full of surprises.

Referee's comment #4: "The authors must provide the pertinent evidence for their assumptions or refer to publications where these evidence are shown. In the absence of such information, it is impossible for any reader to have the smallest idea of the validity of the hypothesis put forward."

Authors' reply #4: The pertinent evidence to justify the existence of a suture zone, which is the aim of the paper, would be, for example, to test positive the juxtaposition of a section of oceanic lithosphere onto a section of continental lithosphere. We think the mafic-ultramafic complex we have mapped in the Mérida Massif is a good example of an ophiolite. But even if serious doubt is held regarding this, it is clear that at least the mafic-ultramafic complex is somewhat equivalent to a lower crust-upper mantle exposure (note also the metamorphism of this unit). Such section rest on top of a tectonic slice with continental crust affinity, i.e. a lower crust-upper mantle section thrust onto continental crust, a process that we can see in a suture zone sensu lato. This can be easily interpreted from the tectonostratigraphy, map and cross-section we are presenting. We are going to include pictures of relevant observations (structures, rock types, etc.) in the revised version of the manuscript so they can be used as a visual reference for the statements we make in the manuscript. All the comments provided here are largely based on a very simple idea: I/we do not believe the geological map you are presenting. Sorry, we cannot fight back against that. Basically because we cannot discuss the reasoning other authors followed when they did their geological maps of the region before, we cannot know which specific outcrops they visited back then, which were the criteria they followed for discriminating between lithologies and, more importantly, the previous maps look like created for a completely different purpose. We are not saying that previous maps are wrong. They were most likely created to show or solve other things. To illustrate this, you could analyze the grouping criteria followed in previous maps. The Serie Negra Group that is presented in some of these maps includes metasedimentary rocks (paragneisses, schists, phyllites, slates), metagranitoids (granites, tonalities, diorites, gabbros) variably transformed into orthogneisses, metavolcanics, serpentinites, and amphibolites, among other things. All those lithologies were mapped as a single unit. There is no way one can identify the relative position between them, as most of them are simply gathered into the Serie Negra Group. There is no way to discriminate whether there is a section that is composed exclusively of some particular type of rock (e.g. mafic-ultramafic ensemble). Moreover, it is quite shocking that in other maps some major faults or shear zones were not recognized (we did), and so on. We are presenting and discussing the criteria we have followed for tracing the boundaries between the units we propose, and we are (briefly) describing the lithologies included in each of the units. It is the least that must be done for presenting a geological map. The result is a new geological map and cross-section for the Mérida Massif, for which a new understanding is presented based on a different approach. Finally, we would not underestimate the capacity of other readers "to have the smallest idea of the validity of the hypothesis put forward", i.e. the existence of a suture zone in the Mérida Massif based on the data and arguments we are presenting. There is plenty of smart, quick-witted geologists out there. We know some.

---

## Referee Comment (RC2) · Manuel Francisco Pereira (Referee) · 20 Jan 2020

This manuscript by Díez-Fernández and co-authors presents a significant dataset obtained from geological (structural) mapping surveys developed in a complex region of the Iberian Variscan belt located in Ossa-Morena/Central Iberian zones boundary. Based on the results obtained, the authors reinterpret the region's structure by defining different tectonostratigraphic units with continental or oceanic affinity. Structural interpretation is plausible and well supported by structural data. My comment is focused on the mafic-ultrabasic rocks of Mérida. I consider that the quality of the discussion could be improved by addressing the following topics: i) It will be useful to confront argu-

ments that consider the mafic-ultrabasic rocks of Mérida as representing a Cadomian island-arc (Bandrés et al., 2002, 2004), instead of oceanic lithosphere (i.e. ophiolite) as you proposed (this study); ii) It will be possible to discuss if the Ediacaran Calzadilla ophiolite (Arenas et al., 2018) and the one now proposed by Díez-Fernández and co-authors (this study), are related in terms of their formation and emplacement; iii) It will be important to clarify whether garnet from the Mérida mafic-ultramafic rocks represent porphyroblasts or/and porphyroclasts, i.e., they grew or not with metamorphism and deformation; this has implications for the interpretation of the Sm-Nd dating obtained on garnet;

---

## Author Comment (AC2) · 26 Jan 2020

Referee's comment #1: "It will be useful to confront arguments that consider the mafic-ultrabasic rocks of Mérida as representing a Cadomian island-arc (Bandrés et al., 2002, 2004), instead of oceanic lithosphere (i.e. ophiolite) as you proposed (this study)"

Authors' reply #1: The interpretation of the Precambrian rocks of the Mérida Massif as a Cadomian island-arc is based on, among other things, the assumption that all of them were part of the same piece of lithosphere. Neither major tectonic contacts were identified within the Precambrian ensemble, nor was even this ensemble divided into tectonostratigraphic units aimed to distinguishing between potentially different geody-

namic settings for each of them. Our proposal tries to build a new understanding for the region from scratch. And for that, we need to build some pillars before introducing a discussion on the geodynamic setting that may explain some specific features of the tectonostratigraphic units. We are setting the basis and thankful for the attention this manuscript is calling. We are not saying that the Cadomian island-arc model does not work for this region, or for some rock ensembles of this region. We just claim our right to start and follow a different path and, maybe, reach a different o more refined conclusions when new data is available and the time comes for it. So far, we prefer to leave that discussion aside since the sole identification of the mafic-ultramafic ensemble as an ophiolite is quite of a new thing for Iberia. The concept ophiolite is not restricted to oceanic lithosphere. This concept has been evolving during the last decades since it was officially coined in the 70's. We must admit that in one sentence of the manuscript we refer to the mafic-ultramafic ensemble of Mérida as a slice of oceanic lithosphere. But everywhere else in the manuscript we are using the terms continental, oceanic, and transitional crust. Nomenclature is important in this regard, as it may lead to misunderstanding. But it is also important to note that we are also using the word "affinity", in order to make it clear that in our opinion the mafic-ultramafic ensemble of the Mérida Massif is something clearly closer to an ocean lithosphere than to a continental lithosphere. Certainly the modern ophiolite concept is quite flexible, and there exist ophiolites related not only to mid-ocean ridges, but also related to other types of marginal basins. We recommend reading literature about the evolution of the ophiolite concept (authors such as Drs. Dilek, Furnes, Pearce, etc. could be a good start).

Referee's comment #2: "It will be possible to discuss if the Ediacaran Calzadilla ophiolite (Arenas et al., 2018) and the one now proposed by Díez-Fernández and coauthors (this study), are related in terms of their formation and emplacement."

Authors' reply #2: The relationship between both ophiolites is a different matter and is far from the scope of the manuscript, which is focused on the recognition, introduction and brief description of tectonostratigraphic units, along with some broad geological

implications. Certainly the manuscript would gain from it, but we need to solve other things before such a discussion is even possible. So, no, such discussion is not possible yet, but it soon will. Not many years ago we started a line of research dealing with suture zone exposures in SW Iberia. We are happy to see this line of research is leading us to some unexpected, maybe revolutionary places. Meaningful and groundbreaking lines of research need time to bloom as they need solid grounds. Step by step, friends, step by step.

Referee's comment #3: "It will be important to clarify whether garnet from the Mérida mafic-ultramafic rocks represent porphyroblasts or/and porphyroclasts, i.e., they grew or not with metamorphism and deformation; this has implications for the interpretation of the Sm-Nd dating obtained on garnet."

Authors' reply #3: In these rocks, garnet is not an orthomagmatic mineral, as it growths onto fabrics formed under solid-state conditions and somewhat implied by former descriptions by Bandrés (2001) and papers derived from his PhD Thesis. In relation to the suggested Sm-Nd isochron, since garnet is not an orthomagmatic mineral, the Sm-Nd dating is not providing an igneous age. Whatever the case, the fabrics in the mafic rocks are Neoproterozoic (Cadomian in a broad sense), so the regional inferences we propose (Variscan vs Cadomian tectonics) would remain the same. This is also supported by further observations made by Bandrés (2001), who recognized that early Cambrian rocks from nearby sectors of the northern Ossa-Morena Complex rest unconformably onto metagranitoids and metasedimentary rocks that exhibit a regional foliation similar to that in the study area. We will add this latter reference to the discussion in the revised manuscript in order to reinforce our conclusions.

We are very thankful for the comments provided by Dr. Pereira, since we know his intention is to make the Geology of Iberia progress. We value and recognize his experience in SW Iberian geology. We are also thankful he is not providing anonymous comments to the manuscript. Courage in science is a rara avis lately.

---

## Referee Comment (RC3) · Anonymous Referee #3 · 4 Feb 2020

The extremely appealing title and nice abstract of this manuscript by Díez Fernández et al. initially caught my attention and invited me to accept its review with the greatest interest. If correct, this topic would attract the interest of many researchers of various disciplines, myself among them! However, I needed several successive readings to finally realize that the only data presented are those making the structural map and sections. Other than these, only interpretations are given from the very beginning, which is really disappointing. If the authors want to prove that an (Neoproterozoic) oceanic affinity unit, and therefore a (Cadomian) suture are present in the Mérida area they should document this with geochemical/isotopic data. Most rocks of their mafic/ultramafic unit have been previously interpreted on the basis of their geochemistry as arc-related

(Bandrés, 2001; Bandrés et al., 2002, 2004). A discussion of this apparent controversy can be only sustained with data, which as said, are missing. In addition, you should not claim for an oceanic suture and not to mention the nature and correlation of the two juxtaposed continental blocks, apart from describing the accretion process. I presume the authors have data of those kinds, otherwise I do not understand how convinced the seem to be of their interpretation. I invite them to enlarge their manuscript and incorporate those data, even if they are thinking in publishing them in a higher rank journal. Without them, this is more an extended abstract, with a nice map and sections, than a paper. I really look forward to seeing the data sustaining your interpretation.

---

## Author Comment (AC3) · 8 Feb 2020

Given the nature of the comment, and others before it, we think it pertinent to provide a general perspective so everyone understand the big picture. The authors are deeply surprised by the nature of this third comment to our manuscript (it is a pity that the person making the comment remained anonymous). Perhaps it is convenient to point out that four out of the five authors of the work are petrologists, geochemists, geochronologists and/or isotopic geochemists. That is, we obviously understand the importance of geochemical data to advance in the understanding of complex orogens. However, we are also Basement Geologists, and we know that all that type of data

is useless (or may lead to misunderstanding) if we do not start by having a very clear idea of the geological context in which they are obtained. We think that doing Petrology-Geochemistry-Geochronology based only on geo-localized samples collected without a fine knowledge of the Regional Geology at hand is a waste of time and money.

To address any work in complex basement regions, it is absolutely necessary to distinguish between lithological units (if any), following consistent criteria. In basement regions classified as "complexes" (it should be remembered that these are regions constituted by different terranes with different origins and tectonothermal evolutions), the recognition of tectonostratigraphic units (or terranes) is something prior to any other data acquisition. This requires a lot of field work, a lot of track runs and a lot of mud in the boots and hours under the sun, rain and cold. That is, it is essential to create good quality maps and geological cross-sections, and also to generate a systematic knowledge of the lithological groups in the area. Only this methodology allows to recognize the existing terranes, according to their lithological coherence, tectonic contacts with other rock groups, contrasting tectonothermal evolution, etc. For us, it has been disappointing that our anonymous reviewer has not been able to perceive the need for all of this to advance in the geology of the Mérida region. In order to objectively assess the contribution represented by the manuscript submitted to this Solid Earth volume, it is convenient to do the quick exercise of comparing the maps and cross-sections previous to those presented here.

At this point, it could be useful to review the history of geological progress in the NW of the Iberian Massif, keeping in mind that this special volume is in honor of Dr. Martínez Catalán. In that region, the geology of the internal zones was first described in modern terms by Parga Pondal and many scientists from Leiden University, who built a regional database based on the recognition of different geological entities defined as "Complexes". This is how the Cabo Ortegal Complex, the Órdenes Complex, the Morais Complex, etc. emerged... It was in 1986 (Arenas, Gil Ibarguchi, González Lodeiro, Klein, Martínez Catalán et al., Hercínica) when the tectonostratigraphic units of these

complexes were collectively described. This step allowed the correlation between complexes, and from there, shortly afterwards, the recognition of the different continental and oceanic domains (ophiolites) that all of them share, thus reaching a point where it was possible to go further and start talking about the terranes involved in the assembly of Pangea from a NW Iberian perspective. This essential methodology for scientific progress in complex basement regions had not been applied until a few years ago in the SW of the Iberian Massif, which in our opinion may have led to considerable confusion. Our research group has been working in the SW for some few years now, following the steps given in the NW decades ago. We have begun to identify ophiolitic units that had gone unnoticed, or had been confused with other types of units. These ophiolites may define a suture or several sutures, we are still working on that, but the recognition of a new ophiolitic unit is, by itself, a contribution that deserves publication in a SCI journal (with all respect to extended abstracts. . .). The discussion about our research increases, which we assume as logical, although we think that the workflow based on: 1) differentiation of tectonostratigraphic units in the SW of the Iberian Massif; 2) correlation to SW scale; 3) SW-NW scale correlation; 4) correlation to the scale of the Variscan Orogen; has arrived here to stay.

The basement geologists have exciting challenges of enormous complexity before us. Constructive collaboration can help us move forward and capture resources for our research lines. We believe that our work is a valuable contribution to the geology of the SW of the Iberian Massif and, in particular, for this volume of Solid Earth, who is dedicated to a scientist who has contributed to significant advances in the geology of the Iberian Massif following approaches similar the one described here and in the manuscript we present. As it is commonly used in the fashionable Soccer World, we ask for a little RESPECT, for everyone, including the honoree.

That being said, in the following paragraphs we will provide specific answers to the comments posed by the reviewer.

Referee's comment #1: "The extremely appealing title and nice abstract of this

manuscript by Díez Fernández et al. initially caught my attention and invited me to accept its review with the greatest interest. If correct, this topic would attract the interest of many researchers of various disciplines, myself among them! However, I needed several successive readings to finally realize that the only data presented are those making the structural map and sections. Other than these, only interpretations are given from the very beginning, which is really disappointing."

Authors' reply #1: We are sorry we did not meet the reviewer's expectations. But it is important to note that here, the reviewer is likely unaware that he/she is contradicting him/herself. He/she says we are presenting data, but a few lines below he/she also says we are not. However, the reviewer, after reading the text several times (as he/she claims) has not realized that there is something else other than the map and cross-sections. This reviewer is apparently missing the brief description and listing of lithological ensembles that we propose. Such a thing took many days of observation and data collection in the field and microscope, as well as an effort towards concision in presentation, but perhaps the reviewer did not think about it.

Referee's comment #2: "If the authors want to prove that an (Neoproterozoic) oceanic affinity unit, and therefore a (Cadomian) suture are present in the Mérida area they should document this with geochemical/isotopic data."

Authors' reply #2: As we already said in a previous reply, geochemical/isotopic data by their own do not prove the oceanic affinity of a unit. It is the recognition of the rock association that works best, and it is what we are presenting here. In other words, you could travel to any of the world-class examples of ophiolites without carrying an ICP-MS in your backpack, and yet you would not hold a doubt about the ophiolitic nature of what you are looking at in the field. Once again we recommend some reading about the ophiolite concept.

Referee's comment #3: "Most rocks of their mafic/ultramafic unit have been previously interpreted on the basis of their geochemistry as arc-related" (Bandrés, 2001; Bandrés

et al., 2002, 2004). A discussion of this apparent controversy can be only sustained with data, which as said, are missing.

Authors' reply #3: The objective of the paper is not discussing the geochemistry and petrological processes involved in the generation of the rocks of the study area. Therefore we do not need to present geochemical data. The main objective of the paper is presenting data that sustain the existence of another ophiolite in SW Iberia, and we do not need geochemistry for that. Remember, you would not carry an ICP-MS in your backpack. . . We do not see any controversy between what we propose and an arc-related setting. We just skip that type of discussion because we are not presenting new data to do it. We think it is an honest approach from our part. Moreover, in the Geological Setting section, we acknowledge such a regional setting for the rocks we study. Perhaps the reviewer should read the manuscript one more time to note this.

Referee's comment #4: "In addition, you should not claim for an oceanic suture and not to mention the nature and correlation of the two juxtaposed continental blocks, apart from describing the accretion process."

Authors' reply #4: There are oceanic sutures of many types, and it is perfectly reasonable to claim for an oceanic suture when you recognize an ophiolite. We are not claiming for any correlation between the continental blocks, we are just identifying them as such, which is more than enough to claim for a suture zone if they are separated by an ophiolite. Funny to read that we are, in a way, allowed to "describing the accretion process". Accretion means a gradual increase or growth by the addition of new layers or parts. From a geological perspective, accretion usually takes place in relation to suturing of oceanic basins (where many ophiolites come from. . .), i.e. in relation to the underthrusting of a lithospheric slab along a subduction zone. In a way, the reviewer is accepting the main idea we are sending with this manuscript.

Referee's comment #5: "I presume the authors have data of those kinds, otherwise I do not understand how convinced the seem to be of their interpretation. I invite them

to enlarge their manuscript and incorporate those data, even if they are thinking in publishing them in a higher rank journal."

Authors' reply #5: Very easy to understand. Our conclusions are independent from geochemistry. Our conclusions rely on basic Structural Geology, basic Igneous and Metamorphic Petrology, basic Tectonics, and a modern understanding of what an ophiolite is.

Referee's comment #6: "Without them, this is more an extended abstract, with a nice map and sections, than a paper."

Authors' reply #6: This sentence (and some others before) is a piece of art. It synthesizes the view of the reviewer about several things. First, the comment despises the value of fieldwork. And more importantly, the value of the data geologists can obtain in the field to support by their own some important interpretations and scientific discoveries. We strongly believe fieldwork is essential and the first step towards more oriented research in Geology. Second, the comment does not recognize a geological map and cross-sections derived from it as data valuable enough to be presented alone in a scientific paper. It seems to us that the reviewer consider this type of data as secondary, as worth it just for a minor purpose. It is sad to remember this here but, a geological map takes many days, weeks and even months of fieldwork to be complete. Not to mention that it requires insight in many regards proper of a scientific contribution. Someone could even say that doing a map may take much longer (and more money and insight...) than analyzing n samples for geochemistry. A geological cross-section is a thorough synthesis in 2D of the more or less complex 3D structure of a region, which also takes time and much experience to be recognized and properly represented. Maybe the reviewer did not think about it but a geological map and derived cross-sections may include (implicitly) more numeric (quantitative) data than many of the analytical tables you can find in a research paper today. Many geological maps have contributed decisively to our understanding about our planet, so please, change your mind. Third, we are not sure if the reviewer is happy with concision in

Science. It seems that there is a minimum length for a manuscript to be considered as a valid scientific paper. Those contributions that do not reach such length would be just extended abstracts, wouldn't they? We recommend the reviewer to contact the Editorial Board of scientific journals such as Geology, Terra Nova, Science, Nature, and many others that (sometimes only) accept quite concise contributions so he/she can explain them what they are doing wrong.

Referee's comment #7: "I really look forward to seeing the data sustaining your interpretation."

Authors' reply #7: You have them already.

---

## Editor Comment (EC1) · Juan Gómez-Barreiro (Editor) · 16 Feb 2020

Authors should present the complete map of lithologies of the area under discussion on which they base their new interpretation. On the other hand, as the concept of ophiolite can be broad, it should be clearly stated/discussed in the text. Besides, authors have to explain what data (petrological, geochemical, structural...) and arguments make authors prefer the interpretation of such units as ophiolites rather than an island-arc.

Juan Gomez Barreiro